# Scoping Review on the Use of Music for Emotion Regulation

**DOI:** 10.3390/bs14090793

**Published:** 2024-09-09

**Authors:** Hyun Ju Chong, Hyeon Joo Kim, Bohyun Kim

**Affiliations:** 1Music Therapy Department of Graduate School, Ewha Womans University, Seoul 03760, Republic of Korea; hju@ewha.ac.kr (H.J.C.); hyeonjookim@ewhain.net (H.J.K.); 2Ewha Music Wellness Research Center, Seoul 03765, Republic of Korea

**Keywords:** music, emotion, regulation, regulation processes, scoping review

## Abstract

With increasing interest in the emotional responses to music, some studies are specifically looking into music’s regulatory function on emotion, known as “music emotion regulation (MER)”. The purpose of this study was to examine the concepts on the regulatory impact of music on emotion using a scoping review procedure. Through an electronic database and manual searches based on the guidelines suggested by the JBI Manual for Evidence Synthesis, a total of 47 studies were identified and included for analysis. The results showed that there were some definitional challenges in each component of music emotion regulation. Most studies treated music as a single variable without exploring the regulatory mechanism of the intra-musical elements that constitute music. When examining the regulatory impact of music on emotion, emotion was inclusive of other terms ranging from feeling to mood. Most of the MER studies employed the terms used in the emotion regulation strategies; however, there were some regulatory processes that pertained solely to music’s cathartic effect. Overall, the results showed that music emotion regulation (MER) was not clearly distinguished from music emotion (ME) studies. Future research should allocate more attention to the theoretical mechanism behind music’s intrinsic regulatory effect in emotion regulation (MER) studies.

## 1. Introduction

Musical behavior is a profound facet of human expression that encapsulates one’s thoughts, emotions, and feelings. By engaging with music as an auditory medium, individuals can communicate and connect with others, experiencing a diverse spectrum of emotions and sentiments. Among various levels of music engagement, receptive music allows listeners to explore emotions triggered by music. The receptive aspect of musical experience enables individuals to grasp and interpret the emotional subtleties conveyed by musical elements, including rhythm, melody, harmony, and lyrics [1].

For decades, researchers have been intrigued by the impact of music on human emotions and its role in regulating them. It is abundantly clear that music serves as a conduit for emotions, prompting investigations into various factors that mediate the relationship between human emotion and music. These factors encompass direct cues, such as psychophysical cues or expectancy mechanisms within the music itself, as well as indirect or subjective sources, such as personal memories and associations [2,3,4].

Over the years, research into the emotional aspects of music has witnessed significant growth, with active exploration dating back to the 1930s [5]. While music is primarily an auditory stimulus perceived through hearing and listening, its effects extend well beyond the auditory domain, evoking multisensory responses at perceptual, cognitive, somatic, and behavioral levels. Despite varying research outcomes that may either support or contradict each other, the evidence unequivocally points to music as a potent elicitor of emotions, driven by the listener’s physiological responses, which stem from the autonomic nervous system to emotional actions influenced by referential experiences. Therefore, when considering the effects of music, it encompasses a diverse range of emotional responses in terms of forms, traits, and dimensions.

Emotion regulation can be defined as purposeful processes aimed at influencing the intensity, duration, and nature of one’s emotional experiences [6]. The term “emotion regulation” is often used interchangeably with “affect regulation” and “mood regulation”. Gross [7] distinguished between emotion and mood based on their degree of behavioral expression and whether they are elicited by specific events. For instance, emotions are typically more behavioral in nature and are often tied to specific events that evoke those emotions. Consequently, the regulation of emotion or moods may target different aspects of emotional response, including physiological responses and behavioral expressions.

The term “regulation” refers to a process aimed at facilitating a transition from one state of being to a more manageable and comfortable one, making emotion regulation a central concept in the field of well-being [8]. However, reaching a comfortable emotional state can involve various pathways, whether it is uplifting the mood, grounding oneself in negative emotions temporarily, or suppressing the emotion [9]. These processes encompass both extrinsic and intrinsic mechanisms for monitoring and adjusting emotional reactions [10]. A meta-analysis of the emotion regulation study indicated that certain strategies—namely attention deployment, cognitive change, and response modulation—were sequentially effective [11].

Furthermore, scientific research has illuminated how music engages multiple regions of the brain, measuring brain activity patterns. With the development of technology, non-invasive neuroimaging devices such as an electroencephalogram (EEG), functional magnetic resonance imaging (fMRI), and functional near-infrared spectroscopy (fNIRS) are used to derive evidence-based effects of music during listening [12,13]. Among the studies using neuroimaging devices, fNIRS studies consistently showed that emotion may have an effect on the construal level, particularly affecting prefrontal cortex (PFC) activation related to the valence and intensity of motivation [14]. These advanced measurement devices are essential for music emotion regulation studies.

In terms of emotion regulation, listeners actively choose music that aligns with their desired emotional outcomes, seeking to shift their mood toward a positive state or alleviate negative emotions [15]. The emotional experience of music manifests across various dimensions, depending on whether the source of the emotional trigger is intrinsic to the music itself (implicit) or arises from the listener’s subjective associations with the music (explicit) [16].

When music is utilized for emotion regulation, it serves as both an explicit and implicit resource. For implicit regulation, music serves as a facilitator for an individual’s emotion regulation process serving as an internal resource, such as inner hearing of any song associated with positive memories or associations. For explicit regulation, music involves a conscious effort to evoke changes in emotional valence, such as listening to uplifting music for affective entrainment. Deliberately selecting music for listening or playing is a common example of using music as an explicit emotion regulation strategy. Bogert et al. [17] demonstrated that different areas of the brain are activated depending on what the listener focuses on during music listening. When they attend to the actual instrumentation of the music piece (implicit processing), regions involved in emotional processing are activated. Conversely, when they concentrate on the emotional content of the music (explicit processing), regions specifically governing emotional processing are activated.

The regulatory process can be approached through either a bottom-up or top-down method, depending on the musical elements employed. In the bottom-up approach, rhythm can be utilized to entrain physiological states, thereby uplifting mood [18,19]. Conversely, in the top-down approach, lyrics or predictable musical progression in the melody can help relax and alleviate distressing states [20,21].

Research supports that rhythm as one of the profound components of music governs arousal level, manifested by the listener’s physiological variable. One of the main principles that support music’s regulatory effect on emotion may be entrainment. Music can entrain physiological states through its rhythmic characteristics [22]. Studies also show that valence and arousal are two constructs that affect each other as well. Further, the combined form of musical components may have the potential to evoke a holistic effect in generating emotions. However, there is a notable scarcity of research on how these musical elements interact synergistically to induce emotional responses.

In the last decades, the field of music emotion regulation has emerged as a distinctive area of study, emphasizing the role of music in regulating mood, enhancing positive emotional states, and reducing negative emotions [15]. In contrast to more established emotion regulation (ER) fields, the field of music emotion regulation (MER) lacks a unified and agreed-upon theoretical framework for examining how music regulates emotions [23].

This study sought to explore the realm of knowledge concerning the regulatory effects of music on emotions, focusing on the use of music to induce emotion regulation in the existing studies on music emotion regulation. To achieve this, a scoping review method was employed. Unlike systematic reviews, which aim to synthesize results for end-users, scoping reviews cover a broader body of literature to clarify evidence in certain areas of knowledge [24]. As music emotion regulation (MER) is becoming a field with practical implications, this scoping review aims to identify how music and its intra-musical constructs are utilized for regulatory effects on human emotion across different literature.

This study aimed to examine the concept of the regulatory effect of music on emotion, including how emotion is defined and what is the rationale behind music selection purported to induce the regulatory effect. The study observed the following five parts pertaining to music emotion regulation studies:1Research trend in MER studies: This part identifies the prevailing trends and patterns in MER studies over time.2Scoping emotion in MER studies: This part explores how emotions are conceptualized and operationalized within the context of MER studies, including specific definitions related to emotional responses to music.3Scoping music and its components in MER studies: This part examines the rationale and working mechanisms behind selected music and its components.4Scoping the regulatory process of emotion in MER studies and domains: This part compares the definitions of regulation between emotion regulation (ER) literature and MER studies and suggests the domains where these regulatory processes occur.5Scoping challenges for components of MER studies: This part summarizes key findings and challenges identified from the scoping review regarding the three components of MER studies, and highlights areas for future research.

## 2. Method

To examine a comprehensive concept of music emotion regulation, the researchers adopted a scoping review methodology. A scoping review, as defined by Arskey and O’Malley [25], aims to map the key concepts that underpin a specific research area. The study purported to explore various concepts and methodological sources that exist in music emotion regulation studies, identifying trends, gaps, and challenges for further investigation.

### 2.1. Step 1: Studies Included for the Review

The scoping review methodology is designed to be inclusive and informative, providing a broad overview of the existing research on the subject. By following these steps, the authors aimed to conduct a comprehensive analysis of the literature related to music, emotion, and regulation. Accordingly, to identify which studies to include, the authors established inclusion and exclusion criteria based on guidelines suggested by the JBI Manual for Evidence Synthesis [26] (Table 1):Keywords: The authors used three main keywords for their search—“music”, “emotion”, and “regulation”—in various combinations using Boolean operations AND to identify relevant literature. These keywords helped narrow the search to articles specifically related to use of music for regulatory effect on music. By using “regulation” as a keyword, the authors distinguished studies focused on music emotion regulation (MER) from broader music emotion (ME) studies, which generally explore a wide range of emotional responses to music.Electronic databases: The authors conducted a search through various electronic databases, including PubMed, PsychINFO, Web of Science, and Scopus. These databases are reputable sources for academic research articles across various fields.Scope of Search: The search was limited to peer-reviewed academic journals published in the English language. This restriction ensures the credibility and quality of the sources.Timeframe: The search covered articles published from 2014 to 2024. This range was chosen to include recent research while avoiding outdated studies.Manual selection: To ensure that all relevant articles were included, the authors manually searched representative journals in the field. These journals include *Musicae Scientiae*, *Psychology of Music*, *Cognition & Emotion*, *Journal of Music Therapy*, and *Nordic Journal of Music Therapy*. This step was crucial for identifying articles that may not be available in the electronic databases.

### 2.2. Step 2: Study Selection

After conducting a search, the authors independently screened studies retrieved to determine whether they met the selection criteria for eligibility. Studies with insufficient information were examined through a full-text review to confirm their inclusion. Three authors evaluated the studies independently according to the predetermined inclusion and exclusion criteria. Subsequently, the authors collaboratively reviewed the studies until a consensus on inclusion was achieved. The flow chart for the article selection process is provided below. Each step sorted the studies, and ultimately, 47 studies were selected (Figure 1).

### 2.3. Step 3: Charting the Data

Based on the exclusion and inclusion criteria, a total of 47 articles were selected for the scoping review. Each article was reviewed and categorized according to specific criteria, including the year of publication, participants (e.g., age, demographics), study design (e.g., experimental, observational), and other relevant characteristics including the following:Music selection and implementation: The authors analyzed the selection criteria of music and rationale for formulating music activities described in each article. This analysis includes identifying the types of music activities used in the studies, such as listening to music, playing musical instruments, singing, or a combination of these activities. They also noted specific details about the musical pieces used for emotion regulation, including duration of music played, number of pieces used, and whether the music was selected by the researcher or the participants.Regulation: The authors analyzed the operational definitions of regulation and how the studies intended to measure music’s regulatory effect. The authors further investigated variables that manifested the changes in the emotion as a regulatory effect.Emotion: The review examined how emotion was defined as a dependent variable. Additionally, other synonyms used interchangeably with “emotion” were examined (Appendix B).

### 2.4. Step 4: Summarizing and Reporting Results

Initially, three authors independently conducted data extraction from the included studies. Subsequently, they iteratively refined and reviewed the identified categories through multiple discussions to ascertain the final agreement to derive the results.

Following a thorough analysis of studies and identifying the scoping theme, the authors used the Preferred Reporting Items for Systematic Reviews and Meta-Analyses extension for Scoping Reviews (PRISMA-ScR) checklist to ensure adherence to the steps and categories of the scoping theme [27]. The PRISMA-ScR checklist provides a recommended guideline for reporting the findings of scoping reviews.

## 3. Results

The study purported to examine the use of music for emotion regulation, as explored in various studies to construct an understanding of music’s regulatory effect on emotion. A scoping review is used in order to identify the knowledge gap, identify concepts, and investigate scoping challenges in conducting research on music emotion regulation (MER) studies. The findings of reviews are presented for each research question.

### 3.1. RQ1: Trends of Music Emotion Regulation Studies

The general characteristics of the 47 selected publications are as follows (Table 2). The trend of studies in music emotion regulation has increased gradually over time. From 2014 to 2016 and between 2017 and 2020, nine studies (19.1%) were published in each period. Most recently, from 2021 to 2024, 29 (61.8%) articles were published, and this significant increase, including the results from the first half of 2024, indicates that research in the field of music emotion regulation is currently very active.

The review showed that the primary music activity for emotion regulation was listening, which accounted for 40 out of 47 studies (85.1%). Following listening, playing the instruments and a combination of various music activities were each implemented in only three studies, representing 6.4% total. Singing was reported in one study.

Regarding the relationship between sample size and experiment setting, studies conducted in experimental environments with between 30 and 99 participants had the highest proportion, accounting for 15 studies (31.9%). On the other side, among the studies that examined emotion regulation through the use of music in everyday life, the largest number of studies (23.4%) involved between 300 and 999 people. In addition, studies often utilized larger sample sizes in both experimental and natural environments. Notably, studies involving more than 1000 participants typically used a specific mobile application to engage with music in everyday life (Table 3). This indicates that while some studies employed experimental designs, a significant portion was conducted in real-world, daily settings.

A total of 47 studies utilized self-report and physiological methods to measure changes in emotion regulation. Forty (85.1%) studies utilized self-report methods, two (4.2%) studies employed physiological measurement methods, and five (10.7%) studies used a combination of self-report and physiological measurement.

Among the studies that used self-report methods, questionnaires and standardized scales were the most common tools. For experience sampling methods (ESMs), a tool to collect real-time data on daily musical experience was used in four studies, and interviews were conducted for qualitative data. For physiological methods, EEG, fMRI, ECG, SCL, EMG, the Face–Word Stroop Task, EDA, and saliva cortisol levels were used for measurement (Table 4).

### 3.2. RQ2: Scoping Emotion in MER Studies

In the 47 selected studies, emotion was typically treated as a dependent variable encompassing a wide spectrum, ranging from somatic responses to behavioral reactions. The term “emotion” was frequently used interchangeably with synonyms or alternative expressions such as mood or affective state and stress response, and they were consistent with terms outlined in [7].

The most frequently used terms were “emotional response or reaction”, accounting for 24 out of 47 studies (51%). “Stress responses or stressful feeling” was used in 11 studies (23.4%), where researchers tried to examine the effect of music on emotion regulation especially when the participants were in a negative situation. “Affects or affective states” and “mood or mood state” were each used in six studies (12.8%) (Table 5).

### 3.3. RQ3: Scoping Music and Its Components in MER Studies

Regarding the selection criteria of music and music engagement time, only 15 (32%) of the reviewed studies mentioned the number of music pieces used or the time of the music engagement. For the rationale/reasons for the selected music activity, 38 (80.9%) studies specified the reasons, referencing existing emotion studies rather than presenting a distinct rationale specific to their own research. On the other hand, the remaining nine studies did not mention any reasons for music activities at all (Table 6).

These findings suggest that there is a lack of a theoretical rationale for musical components and their regulatory effect on emotion. Music for any form of implementation or intervention is particularly difficult to fully describe due to the complexity of music stimuli (e.g., rhythm, pitch, tempo, harmonic structure, timbre). Additionally, it is also challenging to specifically elaborate on the music experiences (e.g., active music making vs. receptive music listening) and other factors unique to its experiential aspects. Consequently, many studies treated music as a single “entity variable”, often neglecting to verify the intra-musical variables involved in their interventions.

Secondly, the duration of music implementation varied tremendously across studies without any specific rationale, allowing listeners subjective control over time, conditions, and the environment, as well as the use of devices (live, recorded, headphones, speakers, etc.). Researchers may have intended to observe the music’s regulatory effect in naturalistic, real-life settings. However, using participants’ personal music pools may weaken the scientific foundation for assessing music’s regulatory effect on emotion, as musical selections vary widely. When the music selection lies in one’s open and subjective music preferences, the study naturally shifts its focus toward examining the regulatory effect from a listener-related orientation rather than a music-related one. In such cases, it becomes crucial to include information not only about the individual’s selected music but also about their motives for choosing and engaging with the music, as these factors contribute to their awareness of any emotional changes. Moreover, a naturalistic environment can also yield confounding variables in the listening condition, such as music being played in the background rather than as a primary focus. This factor can complicate the interpretation of how music directly affects emotional regulation.

### 3.4. RQ4: Defining Regulatory Process of Emotion

The review also examined the operational definitions of regulation used in the context of music emotion regulation (MER), comparing them with those found in the literature on emotion regulation (ER). While most of the terms suggested by the ER literature were adopted by MER studies, demonstrating consistency in the strategic concepts, there were unique terms specific to MER, implying music’s inherent effect. Four clusters of wordings that described the direction of regulation were derived. The first cluster had the most common terms used in MER studies, such as “change”, “modify”, “improve”, “enhance”, “modulate”, “replace”, “alter”, and “adjust”. The second cluster of terms in MER studies was “control”, “manage”, “maintain”, “sustain”, “focus”, and “down-regulate”, implying strategies aimed at holding or containing emotions. The third cluster of terms included expressions like “diversion”, “distraction”, and “expression”, which indicate shifting attention toward alternatives or substitutes. This approach contrasts with strategies that involve containing or holding emotions.

Lastly, the fourth cluster included terms related to the nature of “discharge”, “venting”, “releasing”, and “eliminating” negative affect, connoting a concept of cleansing difficult emotions. Interestingly, these terms differ from those commonly found in the existing literature on emotion regulation that does not involve music. This finding suggests that music possesses a unique mechanism facilitating the release of difficult emotions, thereby leading to a cathartic form of regulation. In other words, music may enable a distinctive process of emotional cleansing, offering a means to alleviate and manage negative emotions in a way that is not typically addressed by other regulatory strategies (Table 7).

The review further examined where and how the regulatory process occurred during or after music engagement. A consensus emerged across many studies that this process is multifaceted, spanning various domains from initial sensation to triggered expressive action or behavior. Specific processes associated with each domain are delineated in Table 7. These domains include physiological, psychological, affective, cognitive, and behavioral aspects. What these findings reveal is the multi-leveled nature of music’s regulatory process, which evolves across various domains, often simultaneously, highlighting the interconnectedness of these dimensions [36].

The findings of the review showed that at the physiological level, music is involved in down-regulating negative emotions, heightened arousal, or uplifting energy levels. At the psychological level, anxiety and stress responses are key variables that are subject to modulation or sedation. In the affective domain, changes in emotions are involved, whether suppressed, induced, or expressed, including coping strategies to improve or stabilize mood. At the cognitive level, the focus is on mental processes such as conscious effort, reappraisal, rumination, or introspection. Webb, Miles, and Sheeran [11] identified this mechanism as particularly effective among emotion regulation strategies. Lastly, the behavioral domain involves activated behaviors such as vocal or physical expression that manifests intense reactions to music. This expressive behavior may involve ventilation or the cathartic discharge of emotions, ultimately leading to a renewed sense of revival [46,81]. This is consistent with the findings of studies that suggest ER is a multi-level process that involves a whole-body emotion system, including subjective feelings, cognition, physiological and neural systems, and behavioral responses [36,37,82] (Table 8).

### 3.5. RQ5: Scoping Challenges for MER Studies

The scoping review findings showed categories of challenges pertaining to the three components of music, emotion, and regulation. Firstly, music as the major medium for regulatory effect, including its intra-musical components, should be thoroughly discussed in the context of music selection criteria. One of the recent fidelity analyses of MER studies showed that the replicability of the experiment was weak due to insufficient information on the music rationale. In order to substantiate well-grounded statements on the logical mechanism of music’s regulatory effect on emotion, enhancing the fidelity and rationale of music’s usage in the MER studies is crucial [23,83]. This challenge includes providing comprehensive details concerning music implementation and data collection methods to study regulatory changes [84,85].

Secondly, given that the primary objective of MER is to bring emotional change, there is a need for a more robust theoretical definition of the term “emotion”. This definition should encompass how emotions are assessed and measured to capture the changes that occur. Furthermore, it is crucial for studies to clarify whether participants are reporting on the emotions they “felt” or those they cognitively appraised or “perceived” in music, as these represent two distinct dimensions (somatic vs. cognitive) of identifying emotions.

Lastly, there must be a clear mechanism linking the selected music intervention to emotion regulation. Despite the studies aiming to induce emotion regulation through music, there was insufficient information regarding the mechanism by which music exerts its regulatory influence. Measuring the regulatory effect presents significant challenges in both ER and MER studies [11]. However, when MER studies specifically target the regulatory impact of music, the procedure must be meticulously designed (Table 9).

## 4. Conclusions

Music emotion regulation (MER) is emerging as a distinct research area that focuses on music as a deliberate and functional tool for modulating and regulating emotions. The scoping review of current studies aimed to explore the use of music for emotion regulation. Music inherently serves as a temporal art of emotion, and it acts as a catalyst and a regulatory medium, augmenting strategies for facilitating emotion regulation. The results indicate minimal distinction between the concepts of music emotion (ME) and music emotion regulation (MER) studies.

Also, deeper contemplation of intra-musical elements that constitute music would be imperative to build a robust theoretical foundation for music’s regulatory mechanism. This would lead to establishing a more compelling mechanism behind music’s “regulatory effect” in order to firmly establish the concept of music emotion regulation (MER) as a unique research area.

The limitation of this study lies in locating all existing literature on music emotion regulation and determining which articles align with the scope of this study. Each article had to undergo a careful review individually for inclusion, even if it contained the target keywords in its title. Often the authors may have had different perceptions regarding the terms associated with them, and therefore, some studies turned out to be irrelevant after the review.

Conclusively, the findings from this study offer exploratory insights and point toward the need for further refinement of the concept of MER for its practical application. While the study provides valuable directionality, it is essential to acknowledge that the field of MER is evolving and being refined, especially the regulatory mechanism. Future research may uncover additional knowledge that was not captured in this initial review.

## Figures and Tables

**Figure 1 behavsci-14-00793-f001:**
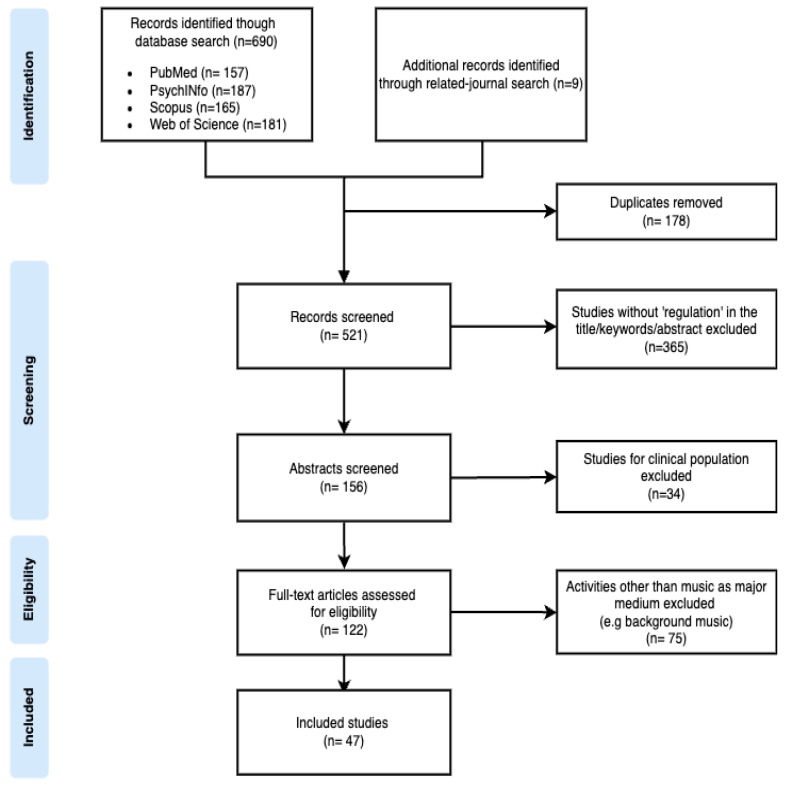
Flow diagram of selecting included studies.

**Table 1 behavsci-14-00793-t001:** Inclusion and exclusion criteria for the scoping review.

Item	Inclusion Criteria	Exclusion Criteria
Subjects	Participants: Non-clinical subjects.	Studies for clinical subjects.
Scope of Keywords	Music: Studies employing various music activities, such as listening, singing, playing, or combined as a primary intervention.	Studies using activities other than music. Studies that used music as an indirect or secondary medium such as meditation, progressive relaxation, etc.
	Regulation: Studies that purported to examine the regulatory effect of music.	Studies on music and emotion in general, without clear purpose of “regulation”. Studies on general emotional response to music.
	Emotion: Studies that aimed to bring changes in emotional realm, including other synonyms such as feeling, mood, affect, etc.	Studies that examined general responses inclusive of emotion, such as quality of life, behavioral management, etc.
Source Types	Studies published in English with quantitative, qualitative, and mixed methodologies.	Studies on the efficacy of music on emotion, such as meta-analysis.

**Table 2 behavsci-14-00793-t002:** Publication year and music activities of studies (*N* = 47).

Factors	Characteristic	No. (%)	Studies
Publication Year	2014–2016	9 (19.1)	[28,29,30,31,32,33,34,35,36]
	2017–2020	9 (19.1)	[37,38,39,40,41,42,43,44,45]
	2021–2024	29 (61.8)	[46,47,48,49,50,51,52,53,54,55,56,57,58,59,60,61,62,63,64,65,66,67,68,69,70,71,72,73,74]
Music Activities	Listening	40 (85.1)	[28,30,31,32,33,34,35,36,37,38,39,40,41,43,44,45,46,47,48,49,50,52,53,55,56,57,58,59,60,63,64,65,66,68,69,70,71,72,73,74]
	Singing	1 (2.1)	[42]
	Playing	3 (6.4)	[61,62,67]
	Combined	3 (6.4)	[29,51,54]

**Table 3 behavsci-14-00793-t003:** Conditions for music engagement: experimental vs. non-experimental condition (*N* = 47).

Music Engagement	Sample Size	No. (%)	Studies
Experiment	Less than 30	2 (4.3)	[58,60]
	30–99	15 (31.9)	[32,36,37,39,41,43,44,47,49,50,59,63,67,71,72]
	100–299	4 (8.5)	[28,45,73,74]
	300–999	1 (2.1)	[57]
	Over 1000	1 (2.1)	[42]
Non-experimental (Daily, etc.)	Less than 30	3 (6.3)	[52,64,70]
	30–99	2 (4.3)	[30,31]
	100–299	6 (12.8)	[34,35,38,46,55,69]
	300–999	11 (23.4)	[29,33,40,48,51,53,56,61,62,66,68]
	Over 1000	2 (4.3)	[54,65]

**Table 4 behavsci-14-00793-t004:** Methods for measuring emotion regulation (*N* = 47).

Measurement		Studies
Self-report (n = 40)	Questionnaire	[28,29,30,31,32,33,34,35,36,37,38,39,40,41,42,43,45,46,48,49,50,51,52,53,54,55,56,57,59,61,62,63,64,65,66,67,68,69,70,71,72,73,74]
	Experience sampling method	[33,64,69,70]
	Interview	[44,60]
Physiological measurement (n = 2)	EEG, fMRI	[28,47,58,59]
	ECG	[36]
	SCL, EMG	[36,37]
	The Face–Word Stroop Task	[59]
	EDA	[37]
	Saliva cortisol	[71]
Combined (n = 5)		[28,36,37,59,71]

EEG: electroencephalograms, fMRI: functional magnetic resonance imaging, ECG: electrocardiography, SCL: skin conductance levels, EMG: facial electromyography, EDA: electrodermal activity.

**Table 5 behavsci-14-00793-t005:** Terms used to define emotion (*N =* 47).

	Keywords	No. (%)	Sample Studies
Emotion	Emotional response or reaction	24 (51)	[30,31,32,33,35,36,39,40,41,42,44,45,46,49,57,58,60,61,62,64,67,71,72,74]
	Mood or mood state	6 (12.8)	[34,50,52,55,70,73]
	Affects or affective states	6 (12.8)	[28,29,38,47,56,69]
	Stress response or stressful feeling	11 (23.4)	[37,43,48,51,53,54,59,63,65,66,68]

**Table 6 behavsci-14-00793-t006:** Information regarding music engagement in MER studies (*N* = 47).

	Characteristics	No. (%)	Sample Studies
Music Engagement Time	Specified	25 (53.2)	[28,32,33,34,36,37,41,43,44,45,47,50,53,57,58,59,62,66,67,69,70,71,72,73,74]
	Not Specified	22 (46.8)	[29,30,31,35,38,39,40,42,46,48,49,51,52,54,55,56,60,61,63,64,65,68]
Number of Music Pieces	Specified	22 (46.8)	[28,31,32,36,37,39,40,43,44,45,47,49,58,59,60,61,62,70,71,73,74]
	Not Specified	25 (53.2)	[29,33,34,35,38,41,42,46,48,50,51,52,53,54,55,56,57,63,64,65,66,67,68,69,72]
Music Engagement Time and Number of Music Pieces	Both Specified	15 (32)	[28,32,36,37,43,44,45,47,58,59,62,70,71,73,74]
	Both Not specified	16 (34)	[29,35,38,40,42,46,48,51,52,54,55,56,63,64,65,68]
	One only	16 (34)	[30,31,33,34,39,41,49,50,53,57,60,61,66,67,69,72]
Reasons for Selected Music Activities	Specified	38 (80.9)	[29,30,31,33,34,35,36,37,38,41,42,43,44,45,46,49,50,52,53,55,56,57,59,60,61,62,63,64,65,66,67,68,69,70,71,72,73,74]
	Not Specified	9 (19.1)	[28,32,39,40,47,48,51,54,58]

**Table 7 behavsci-14-00793-t007:** Key terms defining music’s regulatory function in relation to definitions of emotion regulation (*N* = 47).

Term		Keywords of Music Emotion Regulation (MER) Studies Adopted	Studies
Terms for direction of regulation	Cluster 1: Changing and Modifying Titrate, alter, modulate response [7,75] Coping [76]	Change/modify/improve/enhance/modulate/replace/alter/adjust	[28,33,37,39,41,44,46,48,52,55,64,65,67,70,71]
		Increase arousal/maximize	[37,40,62]
		Modulation of thoughts/affect/behavior	[28,38,61]
		Coping	[53,61]
		Induce/influence	[38,39,51,60]
	Cluster 2: Holding and Maintaining Management of generated emotion [77] Track, assess, control [67] Dampen, maintain, intensify [6]	Control/manage	[49,65,67]
		Maintain/sustain	[28,29,37,46,54,55,64]
		Focus	[14,40,46]
		Reduce/diminish/down-regulate/decrease/minimize	[36,37,52,56,62,68,70]
	Cluster 3: Distracting and Diverging Distraction [78] Diversion [79] Reroute [67] Emotion Expression [80]	Diversion/distracting	[45,56]
		Expression	[39,55,60,62]
	Cluster 4: Discharging and Venting	Discharge/venting/releasing/eliminating	[37,54,56,65]

**Table 8 behavsci-14-00793-t008:** Domains involved in emotion regulation process using music (*N* = 47).

Domain	Regulatory Processes	Sample Studies
Physiological	Down-regulate, heightened arousal, energy	[28,31,36,37,39,40,41,43,47]
Psychological	Modulation of anxiety and stress	[42,48,51,63,68]
Affective	Changes in the generated emotion Improve, control, and maintain mood	[31,32,34,35,36,38,39,41,45,46,47,51,52,53,55,56,57,59,60,66,69,70,71,72,73,74]
Cognitive	Mental process that involves conscious effort, reappraisal, rumination, or introspection	[29,30,31,33,49,52,53,57,58,59,63,67,68,74]
Behavioral	Expressive action or any behavioral expression Intense reactions to goal-oriented changes	[28,31,32,37,48,59,60,61,64]

**Table 9 behavsci-14-00793-t009:** Scoping challenges for each component for further MER studies.

Components	Agendas	Scoping Challenges and Considerations
Emotion	Working definition	Define in relation to other synonyms: feeling, emotion, mood, affect, etc.
	Measurement	Changes in the emotion: perceived (cognitive appraisal) vs. felt (embodied)
	Components/variables	Suggest constituting concepts: valence, arousal, intensity, etc.
Music	Musical elements	Rationale for intra-musical elements: tempo/rhythmic idiom/tonal idiom/other components
	Exposure time and engagement manner	Number of excerpts and duration of each piece
		Specify listening time, condition, situation, etc.
	Selection rationale	Specify researcher’s stance pertaining to music’s facilitative strategy for emotion regulation
Regulation	Regulatory mechanism of music	Provide working rationale and mechanism linking music intervention and emotion

## Data Availability

No new data were created or analyzed in this study. Data sharing was not applicable in this study.

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
