# Peer review of "Scoping Review on the Use of Music for Emotion Regulation"

_behavsci, 2024, doi:10.3390/bs14090793_

Round 1

Reviewer 1 Report (Previous Reviewer 1)

Comments and Suggestions for Authors

This manuscript was previously well-prepared; however, the edits have improved the overall effectiveness of the paper and better represent the study. The inclusion of the table is very helpful, and the researchers included there will further benefit from this manuscript. Congratulations on a well done revision! I look forward to including this article in future work!

Author Response

Reviewer 2 Report (Previous Reviewer 2)

Comments and Suggestions for Authors

Thank you for your attention to the comments and questions that were raised. Overall I believe that you addressed these well, and I appreciate the changes that were made.

I believe there still needs to be some editing of the English language that would make the paper stronger and more readable.

Attention still needs to be paid to the references as the titles of journal articles should not be capitalized. These were not consistent throughout.

Comments on the Quality of English Language

I believe there still needs to be some editing of the English language that would make the paper stronger and more readable.

Author Response

Comments 1: Thank you for your attention to the comments and questions that were raised. Overall I believe that you addressed these well, and I appreciate the changes that were made.

I believe there still needs to be some editing of the English language that would make the paper stronger and more readable.

Attention still needs to be paid to the references as the titles of journal articles should not be capitalized. These were not consistent throughout.

Response 1: Thank you for pointing this out. we have made the modifications accordingly.

p.20-22.

Comments 2: I believe there still needs to be some editing of the English language that would make the paper stronger and more readable.

Response 2: Thank you for pointing this out. We have made some minor word adjustments.

  1. 1 line 17, 20, 22, 31, 43.
  2. 2. line 47, 48, 54, 57, 74, 75, 77, 80, 85- 87, 90-94, 98, 99.
  3. 3 line 102, 103, 113, 115-117, 121, 122, 124, 125, 128-140.
  4. 4 line 153, 156, 157, 159, 165, 166, 170, 171, 175, 176, 181.
  5. 5 line 182, 183, 185-187, 192.
  6. 6 line 204, 209, 210, 222, 223, 225, 226, 229-233, 236.
  7. 7 line 241, 242, 258.
  8. 8 line 290, 298, 299, 304.
  9. 9 line 306-308, 312, 313, 319- 322.
  10. 10 line 336, 337, 352, 353, 362.
  11. 11 line 367-369, 373, 382, 386-388.
  12. 12 line 408, 416, 422, 423, 426, 428.

Reviewer 3 Report (Previous Reviewer 3)

Comments and Suggestions for Authors

Thank you for considering the feedback I suggested. I am comfortable the comments were sufficiently addressed.

Comments on the Quality of English Language

I noted some missing words in the Conclusions, specifically adding "on" prior to "music" (line 409) and "a" before "logical" (line 417).

Author Response

Comments 1: I noted some missing words in the Conclusions, specifically adding "on" prior to "music" (line 409) and "a" before "logical" (line 417).

 Response 1: we have made the modifications accordingly
p. 12 line 408 and 416 (logical --> a robust theoretical).

This manuscript is a resubmission of an earlier submission. The following is a list of the peer review reports and author responses from that submission.

Round 1

Reviewer 1 Report

Comments and Suggestions for Authors

Thank you for the opportunity to read this manuscript. It is extremely clean, very clear, and an excellent contribution to the literature.

The Method is line-in with best practices in scoping reviews and is written beautifully. The writing maintains an academic tone, while being very accessible to readers. It will be of interest to broad readership, and I look forward to including it in future work.

The Results are well-written and reflect reasonable and thoughtful conclusions gleaned from the collected data. These findings corroborate other literature that describes the challenges of having clearly defined definitions and the challenges of music not being explored in the multifaceted way that will be most beneficial for purposes such as emotional regulation and other therapeutic purposes.

My only suggestion would be to include a "Master Article Table" that would list the articles included in the scoping review - maybe it could have a column that designates in which categories the article was included, related to the other tables. For example, article XYZ, (pub year would be in citation), type of study, which music activity, participants, which definition of regulation, etc. This type of chart would serve readers as a quick reference for finding literature and will save readers from having to extrapolate from the references which articles were utilized for which purpose.

Reviewer 2 Report

Comments and Suggestions for Authors

Thank you for this very interesting scoping review. It will definitely add to the literature. The following are my comments, thoughts, questions, and/or suggestions.

1. Please review the manuscript again as there are several places that small words like “a” or “the” should be added.

2. Page 2, line 88 – what is meant by “shorted”?

3. Page 2, line 95 – are you referring to your study? If so, you may want to say “this study”. It might help clarify that it is your study. Also in the first sentence of the paragraph you said “aimed”, and in the second sentence you said “aims” – please be consistent with tense. You probably want to use past tense throughout the whole document since the study has been completed.

4. Page 2, line 98 – “rationale” is used twice close together – see if there is another word you can use one of the times.

5. Page 2, line 100 – states “four parts”; however, on page 3, five parts are listed. Please clarify.

6. Page 3, lines 103 and 108 you may want to change “the” to “this” to be consistent with the other 3 points.

7. Methods – would you please clarify who found the studies? Was a program such as Covidence or Rayyan used to search and manage them? Did a medical librarian help with the search?

8. Page 5 - Excellent flow chart. In the third bubble on the right, I think perhaps you mean “studies”, not “studied”?

9.  Page 6, lines 187-189 – what is meant by “if the nature of information on the studies are too revealing of the authors, they are not mentioned in the reporting”?

10. Page 8, lines 264-266 – I understand your thoughts about participants’ personal music pools potentially weakening the scientific foundation for understanding music’s regulatory effect on emotion due the variance in the musical attributes; however, it is this variance that can bring about emotion regulation. Music preference is different for everyone, and research has demonstrated that an individual’s preferred music is what will be the most effective for them (for example, for some people heavy metal music actually relaxes and calms them, despite it’s fast and strong rhythm and loud volume). Therefore, would it not be important to investigate participants’ use of their preferred music?

11. References – please make sure the capitalization of journal names is consistent throughout. There are also some periods missing after the last initial before the dates.

Comments on the Quality of English Language

I think overall this is very good. It just needs to be reviewed for missing small words here and there such as "a" and "the".

Reviewer 3 Report

Comments and Suggestions for Authors

This manuscript reports findings from a scoping review aimed at examining published research on music and emotion regulation to highlight key characteristics and gaps in the literature. The authors extracted data from 45 research articles that met inclusion criteria and reported findings for how the concepts of “music,” “emotion,” and “regulation” were defined, as well as noted challenges in the current literature. I commend the authors for conducted the study, which seemed to adhere well to appropriate scoping review methods. However, I have some concerns that 

First, what is missing in the introduction/lit review is a rationale for why this study is needed. This rationale should address the gaps in the literature on this topic area, which inform the study aims and research questions, and lead to why a scoping review is the most appropriate methodology. There is existing literature on the topic of music and emotion regulation, including other systematic reviews (e.g., Cook et al., 2019; Uhlig et al., 2013), so what gaps in current knowledge is this project aiming to address?

I recommend being careful with how certain concepts are defined and described. First, although there is a robust area of research in music and emotion regulation, I am not aware that “music emotion regulation” is a distinct field as claimed in the abstract. Additionally, I encourage the authors to be careful about framing emotion regulation as shifting towards “a more manageable and comfortable” emotional state (p 2 line 63). ER is about having the ability to manage and shift our emotional experiences and their expressions, regardless of the direction (increase, decrease, or maintain). This does not necessarily mean promoting positive emotions and alleviating negative ones (p 2, line 78-79); rather, it depends on a person’s goal. For example, sometimes the goal is to continue to feel sad, so that we can be in the space and work through the feeling. Finally, I encourage the authors to be careful about solely attributing rhythm and tonal components to implicit and explicit ER, respectively. There is so much more to how music can impact how we regulate our emotions, it is not as simply as what is described on p 2, lines 83-94.

Related to the concept of music, I encourage the authors to be clear in their definitions, data extraction, and reporting on the distinction between music (as a stimulus), music experiences (e.g., listening), and music intervention (which involves a clinical application, such as with music therapy or music medicine). It does not seem from this manuscript that the authors were in fact including music intervention studies, especially given that clinical populations were excluded, so use of the term “intervention” seems misleading. Relatedly, the authors may find useful the guidelines for reporting music-based interventions published by Robb, Burns, & Carpenter (2011). Although the focus is not on interventions in this project, these guidelines may provide a useful framework for more clearly identifying and reporting important components related to “music” as reported in the literature. 

In the Results section, first paragraph of RQ 3 (p 8) the authors concluded that the music selection was not rigorous addressed in the 30 studies in which participants selected the music. I encourage the authors to re-think this conclusion. Rigor does not necessarily equate with who selected the music; what is more important from a rigor perspective is that who selected the music is clearly and transparently reported, as is the reason for this decision and what music was selected.

Also in the Results section, some clarifications are needed in reporting findings for RQ 5 (p 10). First, the RQ as written on p 3 mention highlighting key findings and insights; this is missing from the results which focus solely on challenges. Two, the authors write that there is a need for research to include more “well-defined theoretical rationales,” yet that information is not reported in the findings.

Other more minor issues include:

- A scoping review is a stand-alone type of research study; although it is systematically conducted, it is not called a “systematic scoping review”

- Recommend adding 1-2 statements in the abstract on the methodology.

- Citations needed p 1 line 40; p 2, line 87

- p 2 line 100 authors write there are four parts explored, yet identify 5 on the next page

- p 4 Table 1, other than a meta-analysis, were other systematic reviews 

- p 7 lines 233-235, I do not understand the connection between the “discovery” and how that aligns with the emotion definition concept.

- p 9 lines 287-288, it is not clear to me what the authors mean that there is a “lack of differentiated concept pertaining to music”

- p 9 line 291, the authors mention “measurement tools employed” but do not report what measurement tools were employed in the data set. Please add this.

Comments on the Quality of English Language

I recommend the authors request or seek out a careful read through by a native English speaker. Overall this manuscript is clearly and well-written; however, there are times where words such as “a” or “the” seem to be missing (e.g., abstract lines 8 and 11) as well as some oddly worded sentences (e.g. p 2 line 70-74, p 2 line 97, p 6 lines 187-188) and words used (e.g., “shorted” p 2 line 88).

Round 2

Reviewer 3 Report

Comments and Suggestions for Authors

Thank you for the careful consideration and incorporation of feedback in this revised manuscript. I find this version much improved, in particular in clarifying the concepts and providing a rationale for why this study is needed. I feel my comments were sufficiently addressed.

Congratulations on the manuscript.

Comments on the Quality of English Language

I have two minor items from reading this revised manuscript: (1) In some of the newly added text I noted some anthropomorphism, e.g. p 8 line 288, p 9 line 309; (2) pg 11 line 351 the sentence started with “Since music’s intrinsic trait…” reads oddly to me. I recommend revising for clarity.
